# One Year after Mild COVID-19: The Majority of Patients Maintain Specific Immunity, But One in Four Still Suffer from Long-Term Symptoms

**DOI:** 10.3390/jcm10153305

**Published:** 2021-07-27

**Authors:** Andreas Rank, Athanasia Tzortzini, Elisabeth Kling, Christoph Schmid, Rainer Claus, Eva Löll, Roswitha Burger, Christoph Römmele, Christine Dhillon, Katharina Müller, Philipp Girl, Reinhard Hoffmann, Stefanie Grützner, Kevin M. Dennehy

**Affiliations:** 1Department of Hematology and Oncology, Medical Faculty, University of Augsburg, 86156 Augsburg, Germany; athanasia.tzortzini@uk-augsburg.de (A.T.); christoph.schmid@uk-augsburg.de (C.S.); rainer.claus@uk-augsburg.de (R.C.); 2Institute for Laboratory Medicine and Microbiology, Medical Faculty, University of Augsburg, 86156 Augsburg, Germany; elisabeth.kling@uk-augsburg.de (E.K.); eva.loell@uk-augsburg.de (E.L.); reinhard.hoffmann@uk-augsburg.de (R.H.); Kevin.Dennehy@uk-augsburg.de (K.M.D.); 3Institute for Transfusion Medicine and Haemostasis, Medical Faculty, University of Augsburg, 86156 Augsburg, Germany; roswitha.burger@uk-augsburg.de (R.B.); stefanie.gruetzner@uk-augsburg.de (S.G.); 4Department of Gastroenterology and Infectious Diseases, Medical Faculty, University of Augsburg, 86156 Augsburg, Germany; christoph.roemmele@uk-augsburg.de; 5Department of Pathology, Medical Faculty, University of Augsburg, 86156 Augsburg, Germany; christine.dhillon@uk-augsburg.de; 6Bundeswehr Institute of Microbiology, 80937 Munich, Germany; katharina5Mueller@bundeswehr.org (K.M.); PhilippGirl@bundeswehr.org (P.G.)

**Keywords:** COVID-19, SARS-CoV-2, persistent symptoms, humoral immunity, cellular immunity

## Abstract

After COVID-19, some patients develop long-term symptoms. Whether such symptoms correlate with immune responses, and how long immunity persists, is not yet clear. This study focused on mild COVID-19 and investigated correlations of immunity with persistent symptoms and immune longevity. Persistent complications, including headache, concentration difficulties and loss of smell/taste, were reported by 51 of 83 (61%) participants and decreased over time to 28% one year after COVID-19. Specific IgA and IgG antibodies were detectable in 78% and 66% of participants, respectively, at a 12-month follow-up. Median antibody levels decreased by approximately 50% within the first 6 months but remained stable up to 12 months. Neutralizing antibodies could be found in 50% of participants; specific INFgamma-producing T-cells were present in two thirds one year after COVID-19. Activation-induced marker assays identified specific T-helper cells and central memory T-cells in 80% of participants at a 12-month follow-up. In correlative analyses, older age and a longer duration of the acute phase of COVID-19 were associated with higher humoral and T-cell responses. A weak correlation between long-term loss of taste/smell and low IgA levels was found at early time points. These data indicate a long-lasting immunological memory against SARS-CoV-2 after mild COVID-19.

## 1. Introduction

Since the end of 2019, SARS-CoV-2 spread worldwide and caused a pandemic [1]. In the acute phase of coronavirus disease 2019 (COVID-19), affected humans usually suffer from mild symptoms similar to a flu-like infection; however, elderly people especially also develop pneumonia, leading to sepsis with lethal outcomes [2,3]. In contrast to many other viral infections, convalescents relatively often suffer from prolonged symptoms (“long COVID-19”). The most common symptoms are a persistent loss of taste and smell, headaches, concentration disorders and fatigue [4]. Other complaints include respiratory impairment, especially in patients with a severe COVID-19 course, and muscle or limb pain [5]. These symptoms post-COVID-19 decrease in frequency over months after the acute phase of infection [6], but how long they may persist, and whether they correlate with specific immunity, is unclear up to now.

Another unanswered question is how long after COVID-19 does immunity protect against re-infection. Some reports of recurrences of COVID-19 symptoms within weeks of infection have been published, but these appear to be isolated cases [7]. Nevertheless, it must be assumed that, especially in patients with asymptomatic or mild COVID-19, re-infection is possible, due to the disappearance of specific anti-SARS-CoV-2 antibodies and T-cells within a few months after primary infection. This could be prevented if individuals are vaccinated before a critical drop in immunity occurs. Most published data of immune memory against SARS-CoV-2 show a more or less consistent decrease in IgA, IgG and neutralizing antibody levels as well as a decline of specific CD4+ and CD8+ cells within the first 100 to 200 days after COVID-19 [8]. Up to now, it is not clear whether the loss of humoral and cellular immunity is continuous or whether there is at least a gradual stabilization of remaining antibodies and T-cells during the first year after COVID-19.

In this prospective study, we focused on COVID-19 patients with a mild course, which is most common in adults at approximately 80% [9]. Such patients were questioned regarding persisting symptoms or secondary disorders using a standardized questionnaire one year after their acute disease. Additionally, humoral as well as cellular immunity against SARS-CoV-2 were measured 6 and 12 months after COVID-19.

## 2. Materials and Methods

Convalescent plasma donors in the Institute for Transfusion Medicine, University Hospital Augsburg, who experienced mild COVID-19 according to the WHO classification [10], were prescreened for study inclusion. Data on medical history were obtained from the plasma donor file. The study was approved by the ethical review board of Ludwig-Maximilian-University of Munich. Signed informed consent was obtained from all participants. Patients were included in the study after recovery from COVID-19. A first analysis of humoral immunity against SARS-CoV-2 by IgA/IgG ELISA was carried out as part of 6-week follow-up visit after onset of COVID-19 and repeated at a 6-month and 12-month follow-up, respectively. IgG and IgA antibodies directed against S1 protein were analyzed using the in vitro SARS-CoV-2 ELISA assay (Euroimmun, Lübeck, Germany). Test results are given as the ratio of binding of antibodies in tested plasma to that of a calibrator (cut-off index values). A ratio > 1.1 indicates positive results.

Additionally, neutralizing antibodies (Nab) were measured at 6-week and 12-month follow-up as described by Haselmann et al. [11]. SARS-CoV-2 (strain MUC IMB-1) was cultured in Vero E6 cells. Virus stocks (50 TCID/50 µL) were prepared and stored at −80 °C until further use. First, serum samples were diluted in 96-well culture plates (Greiner bio-one, Frickenhausen, Germany) in Minimal Essential Medium (MEM, plus non-essential amino acids solution and antibiotic–antimycotic solution; all Invitrogen, ThermoFisher Scientific, Darmstadt, Germany) starting with a ratio of 1:5 up to a maximum of 1:640. Virus was then added to each well and the serum–virus solution was incubated for one hour at 37 °C (5% CO_2_). Afterwards, Vero E6 cells (1 × 10^4^ cells/50 µL) were added to each well and incubated for another 72 h at 37 °C (5% CO_2_). Then, the supernatants were discarded and the wells were fixed in 13% formalin/PBS and stained with crystal violet. Both known positive and negative serum samples were used as controls along with a mock control and a back-titration of virus on each plate.

To capture cellular immunity against SARS-CoV-2, INFgamma (INFg)- and interleukin-2 (Il2)-ELISpot assays were used at 6- and 12-month follow-up. Ex vivo ELISpot/FLUOROspot assays were performed using the Interferon-γ (IFNγ) and interleukin-2 (IL-2) CoV-iSpot kit from Autoimmun Diagnostika (AID GmbH, Straßberg, Germany). Peripheral blood mononuclear cells (PBMC) from citrate blood drawn in the late afternoon were isolated the following morning through Ficoll-Paque (GE Healthcare, Chicago, IL, USA), and seeded in duplicate at a density of 2 × 10^5^ cells per well in AIM-V medium. Cells were stimulated for 18 h with the AID GmbH CoV-2 peptide library containing peptides from the S, N, M and O proteins and CD28 antibody, or left unstimulated with CD28 antibody alone as a negative control. As further controls to confirm specific responses to SARS CoV-2 peptides, we stimulated cells in parallel with peptide libraries covering all coronaviruses (PAN), as well as cytomegalovirus, Epstein–Barr virus and influenza viruses (CEF). Pokeweed mitogen was used as a positive control. Spot numbers were counted using the AID GmbH iSpot Reader. Samples were excluded if the pokeweed mitogen positive control was less than 50 SFU/2 × 10^5^ cells, or if the unstimulated control values were >10 or >20 SFU/2 × 10^5^ cells for IFNγ and IL-2, respectively. Positive responses were stringently defined, essentially as described by AID GmbH, as samples that had a stimulation index of ≥7 for unstimulated control values ≤2 SFU/2 × 10^5^ cells, and a stimulation index of >3 for unstimulated control values >2 SFU/2 × 10^5^ cells. We used a cohort of 32 seronegative platelet donors as negative controls for the assay.

In addition, SARS-CoV-2-specific CD4+T-helper cells (THC, CD25hi CD134hi) were identified using activation-induced marker (AIM) assays by flow cytometry as described by Reiss et al. [12]. PBMC used in ELISpot assays were plated in round-bottom 96-well plates at a density of 1 × 10^6^/well in AIM-V medium, and incubated with medium alone as an unstimulated control, or with the AID GmbH CoV-2 peptide library containing peptides from the S, N, M and O proteins for 18 h. Cells were blocked with TruStain (BioLegend, San Diego, CA, USA) for 30 min, before staining with CD4-Pacific Blue, CD134-FITC, CD25-PEdazzle, CCR7-PerCP-Cy5.5, CD45RO-APC-Cy7, CXCR3-PE-Cy7 and CCR6-APC (BioLegend) for 60min at 4oC. Cells were then washed with PBS and analyzed on a Cytoflex S cytometer, with gating on live cells in the FSC/SSC gate, as shown in the gating strategy (Figure 1). For analysis, both background subtracted (CD25hi CD134hi stimulated cells minus CD25hi CD134hi unstimulated cells as % total CD4+T-cells) and stimulation indices (CD25hi CD134hi stimulated cells divided by CD25hi CD134hi unstimulated cells as % total CD4+T-cells, where any zero count values were set to the lower limit of 0.01% total CD4+T-cells) were calculated. A stimulation index of greater than 2 indicated CoV-2-specific T-cells. In participants with detectable specific THC, proportion of center memory cells (CD45RO+ CCR7+; Tcm), and effector memory cells (CD45RO+ CCR7-; Tem) were determined, and the ratio of Tem to Tcm was calculated.

Long-term symptoms/secondary disorders were recorded by a structured questionnaire (Appendix A) from study participants at 12-month follow-up. Pre-planned correlative analyses were preformed between humoral and cellular immunity assays. Furthermore, the impact of participant’s age and gender, occurrence of fever during acute infection and disease duration, as well as persistent symptoms to performed antibody respective T-cell assays, were analyzed.

### Statistics

Frequencies of persistent symptoms recorded from the questionnaire were calculated for time periods of ≥1, 3, 6 and 12 months after onset of COVID-19. Wilcoxon test (two-tailed) was used to detect significant differences in ELISpot stimulation indices and SARS-CoV-2 antibody values, as well as for pre-planned correlative analyses. Correlations between IgA/IgG antibody values and titer of neutralizing antibodies or T-cell response were estimated by the Spearman test for non-normally distributed data.

## 3. Results

Of the 107 plasma donors approached, 83 included study participants attended all three scheduled follow-up visits and were evaluable for the study. A consort diagram of study participation is shown in Figure 2. Participant characteristics and data regarding the acute phase of COVID-19 are shown in Table 1. Six participants were vaccinated before the 12-month follow-up (five at least 2 weeks before and one just 3 days before follow-up); no participant was vaccinated at the 6-week or 6-month follow-up. At the 12-month follow-up, the subgroup of five vaccinated participants (at least 2 weeks before follow-up) included two females and three males with a median age of 27 years (range: 20–47), while the subgroup of non-vaccinated participants contained 18 females and 59 males (age: 42 years (19–62)).

### 3.1. Clinical Follow-Up

51 (61%) study participants complained of longer-lasting symptoms after the acute infection symptoms had resolved, which decreased over time, but were still reported by 23 (28%) participants after one year. The most frequent complaints were loss of smell or taste, concentration difficulties and headache. Disorders reported at lower frequencies included dyspnea upon exertion and muscle, back or thoracic pain (Figure 3A, Appendix A). Most participants reported only one long-term complication, whereas loss of smell and taste or back and muscle pain were reported also in combination (Figure 3B–E). Nine (11%) study participants developed a cold or febrile infection, and new-onset disease of hair loss, arterial hypertension and anal venous thrombosis affected one participant each.

On the visual scale of the questionnaire, 56 (67%) participants reported a complete restoration of resilience and fitness compared to the period before COVID-19 at the 12-month follow-up. Minor limitations (scale-1) were reported by 16 participants (19%), and mild (scale-2), moderate (scale-3) or severe limitations (scale-4) were reported by 5 (6%), 2 (2%), and 1(1%) participants, respectively. Three (3%) participants reported having a better fitness than before COVID-19. No participant reported very severe limitations (scale-5). The occurrence of persistent symptoms or impaired fitness did not correlate with age, sex or fever during the acute phase of COVID-19.

### 3.2. Humoral Immunity

At the 6-week follow-up, a high proportion of participants had specific IgA and IgG (76/83 (92%) and 79/83 (95%), respectively) antibodies, which decreased to 65/83 (78%) and 58/83 (70%), respectively, at the 6-month follow-up, and further decreased to 60/77 (78%) and 55/77 (66%), respectively, at the 12-month follow-up (only non-vaccinated participants were considered at the last visit). Median levels of IgA and IgG antibodies decreased from 3.4 and 3.7, respectively, at 6 weeks, to 1.8 and 1.8 at 6 months, to 1.9 and 1.7 at 12 months (Table 2, Figure 4A,B for non-vaccinated patients). There were high correlations between IgA and IgG at the 6-week (R^2^ linear = 0.313, correlation coefficient (cc) = 0.545, *p* < 0.001), 6-month (R^2^ linear = 0.413, correlation coefficient (cc) = 0.640, *p* < 0.001), and 12-month follow-up (R^2^ linear = 0.303, correlation coefficient (cc) = 0.592, *p* < 0.001, non-vaccinated participants only, Figure 5A).

Neutralizing antibodies (titer of 1:5 or higher) could be detected in 66/83 (80%) participants of the total study cohort at the 6-week follow-up, and in 37/77 (48%) non-vaccinated participants at the 12-month follow-up. Nab correlated with IgA and IgG antibody levels at both the 6-week (R^2^ linear = 0.202, cc = 0.457, *p* < 0.001 and R^2^ linear = 0.603, cc = 0.775, *p* < 0.001, respectively) and 12-month follow-up (R^2^ linear = 0.163, cc = 0.432, *p* < 0.001 and R^2^ linear = 0.541, cc = 0.742, *p* < 0.001, respectively, Figure 5B,C).

Vaccinated participants had very high median levels of IgA (22.3) and IgG (35.0) values as well as Nab (1:320) at the 12-month follow-up (Table 2).

### 3.3. Cellular Immunity

Of the evaluable participants at the 6-month follow-up, 40/51 (78%) had T-cells that produced INFg, and 24/32 (75%) had T-cells that produced IL-2, as measured by the ELISpot assay. A lower proportion of non-vaccinated evaluable participants had T-cells that produced INFg (48/76 (63%) and IL-2 (30/70 (43%)) at the 12-month follow-up. Only one of the 32 CoV2-seronegative platelet donors (INFg: median: 0.5 SI (range: 0.0–12.0), and Il-2: 2.5 SI (0.5–18.5)), which we used as negative controls for the assay, had a positive T-cell response, possibly due to T-cell cross-reactivity between coronaviruses as previously described [13,14]. Values of stimulation indices are shown in Figure 4 and listed in Table 3.

Specific THC could be detected in 56/70 (80%) non-vaccinated evaluable participants by the AIM assay at the 12-month follow-up. The THC population in non-vaccinated participants contained 39% central memory T-cells and 58% effector memory T-cells with a Tem/Tcm ratio of 1.5 (Table 3).

The proportions of THC measured by flow cytometry showed a high correlation with the stimulation index values of the INFg Elispot assay (R^2^ linear = 0.252, cc = 0.420, *p* < 0.001, Figure 5D).

### 3.4. Correlative Analyses

High correlations could be found between IgG antibody and neutralizing antibody values with specific T-cells measured by INFg- and IL-2-ELISpot assays as well as THC detected by the AIM assay (R^2^, cc and *p*-values are listed in Appendix A, see also Figure 5E,F) in non-vaccinated participants at the 12-month follow-up.

Sex of participants had an impact of median IgA antibody levels at the 6-week follow-up, with lower values in females (1.8 (range: 0.9–10.0) vs. 3.6 (0.8–11.0); *p* = 0.005). This sex-dependent difference was not seen at the 6-month or 12-month follow-up.

Age influenced cellular immunity measured at the 12-month follow-up. Older (≥median age of 42 years) non-vaccinated participants had higher stimulation indices (SI) in INFg- and IL-2-ELISpot assays (9.5 (range: 0.0–93) vs. 5.2 (1.9–53), *p* = 0.006, respectively, (3.5 (0.7–25) vs. 2.5 (0.9–29), *p* = 0.031), as well as a higher SI in the double-positive IFN+IL-2+ ELISpot assay (4.5 (range: 0.0–29) vs. 3.0 (0.0–12), *p* = 0.043). This difference was also seen in the PAN control INFg Elispot assay (8.0 (1.0–68) vs. 4.6 (0.0–52), *p* = 0.014). Furthermore, older participants had a higher proportion of specific THC in the AIM assay (0.17% (0.01–1.71) vs. 0.06% (0.01–1.20, *p* = 0.003).

A longer duration of the acute phase of COVID-19 (>median duration of 10 days) was associated with a higher IgG antibody level at the 6- and 12-month follow-up (2.7 (0.2–6.3) vs. 1.3 (0.4–7.5), *p* = 0.043, and 3.3 (0.6–9.5) vs. 1.4 (0.6–10.6), *p* = 0.024, respectively; non-vaccinated only). The duration of the acute phase of COVID-19 also correlated with the SI of the double-positive IFNg+IL-2+ ELISpot assay (R^2^ linear = 0.170, cc = 0.265, *p* = 0.027), with significantly more double-positive T-cells at the 12-month follow-up in participants with a longer versus shorter acute phase of COVID-19 (12/31 (39%) vs. 3/39 (8%); *p* = 0.002).

Loss of smell or taste up to 3 months was associated with lower IgA levels as measured at both the 6-week and 6-month follow-up (1.9 (0.8–9.7) vs. 3.6 (0.8–11), *p* = 0.016, and 1.4 (0.4–6.0) vs. 2.1 (0.4–9), *p* = 0.006, respectively) but not at the 12-month follow-up.

All other participants’ demographics and characteristics, as well as long-term symptoms, did not influence humoral or cellular immunity in our study cohort.

## 4. Discussion

In this study, we report long-term clinical and immunological data from patients up to one year after mild COVID-19. The proportion of participants complaining about persistent symptoms decreased over time but was still surprisingly high at 28% after 12 months. We found no correlations of specific immunity with long-term complications, apart from a weak association of low IgA at early time points in patients with loss of taste and/or smell. We show that the duration of acute infection correlates with humoral and T-cell immunity. While most participants still have specific antibodies, only half have neutralizing antibodies after 12 months. By contrast, around two-thirds of participants maintained IFNg+-specific T-cell responses at the 12-month follow-up, and 80% of participants had specific T-helper cells, including long-lived central memory T-cells, indicative of long-term T-cell immunity.

The most frequent long-term symptoms were neurological: most frequently the loss of smell and/or taste followed by a headache and difficulties with concentration. This finding is consistent with published data that also show that neurologic symptoms most often persist [15]. In particular, a persistent loss of taste and smell are very commonly associated [16], as also shown in our study.

The most common physical limitation was dyspnea upon exertion, which was still persistent in 8% one year after COVID-19. This long-term complication after one year is remarkable because all participants in our study had only a mild course of disease and almost none were hospitalized. Such clinically relevant long-term complaints in out-patients were also reported by other research groups [17], suggesting a lung involvement in this patient cohort, even though this had not been diagnosed by imaging during the acute phase of the disease. Within the first year after COVID-19, only 11% of participants reported a cold or febrile infection, suggesting that SARS-CoV-2 infection does not cause long-term immunodeficiency. The concern about immunodeficiency following COVID-19 is due to the reduced lymphocyte count during the acute infection [18], as has been observed in some other viral infections [19,20,21]. A low frequency of subsequent infections in our cohort, together with follow-up data showing normal lymphocyte counts and normal lymphocyte subset distribution one month after mild COVID-19, suggest that any possible immunodeficiency would be short-lived [22]. Other persistent physical complaints or new-onset conditions were rare and reported only in individual cases.

In order to capture the symptom of fatigue, which is not easy to detect by medical history, study participants were asked about their restoration of resilience and fitness on the visual scale of the questionnaire. Only two-thirds reported a completely restored health, whereas one third still complained about restricted fitness one year after mild COVID-19.

Levels of IgA and IgG antibodies at the 6-month follow-up were approximately half that of their assumed maximum value measured 6 weeks after disease onset [23], and remained stable up to the 12-months follow-up, thereby showing a biphasic decrease as previously described [24,25]. In contrast to IgM antibodies, IgG antibodies specifically directed against the spike protein of SARS-CoV-2 persist relatively stable over time as described by Dan JM et al., who found a percentage of 90% spike IgG-positive convalescents half a year after COVID-19 [26]. However, similar antibody decreases with persistent sero-positivity, as in our analysis in patients with mild or moderate COVID-19, have also been reported by other investigators [27,28], but with somewhat shorter follow-ups of 7 and 4 months, respectively. Despite these pronounced decreasing antibody levels, humoral immunity to SARS-CoV-2 was demonstrated in nearly 80% of participants in our study participants at the 12-month follow-up. Neutralizing antibodies, which are often used as biological markers of humoral protective immunity [29], also decreased from 80% 6 weeks after infection to 50% one year later. A previous study similarly showed that health care workers all developed Nab three weeks after a mild course of COVID-19, but 15% of those studied had lost detectable Nab [30] three months later, suggesting that they are at risk of reinfection relatively soon after primary infection.

CoV-2-specific T-cells in the participants of our study showed a marked decrease in T-cell responses measured by INFg-Elispot from 78% at 6 months post-infection to 63% at 12 months post-infection. There was also a significant decrease in the absolute levels of IFNg and a strong decrease in IL-2 levels. Given that ELISpot predominantly measures immediate responses from effector and effector memory T-cells, this assay may not be suitable to measure the longevity of T-cell responses. For this reason, we additionally used AIM assays to measure the total CD4+ helper, central memory and effector memory T-cell responses. Specific T-helper cells were found in 80% of participants one year after mild COVID-19, with prominent central memory T-cell components, indicating that the majority of patients even with mild COVID-19 develop memory T-cell responses required for long-term protection from severe disease [31]. This assumption is supported by an observation from Wuhan, China, where the majority of COVID-19 patients also showed a decrease in T-cells directed against SARS-CoV-2 in a biphasic manner. Comparable to our study, the Chinese study found a loss of specific CD8 T-cells in 26% of convalescents and a loss of specific CD4 helper cells in only 16% after 9 months [32].

Considering possible factors influencing specific immunity to SARS-CoV-2, we found positive correlations between the duration of acute illness and the level of IgG antibodies, as well as double-positive IFNg+IL-2+-specific T-cells, in our study. This indicates that patients with a short term of COVID-19 probably have a higher risk to lose specific immunity against SARS-CoV-2 within months. Age also had an impact, particularly on cellular immunity with higher levels of specific T-cells in older participants than younger ones, as measured by the AIM assay. Surprisingly, older participants responded significantly better to epitopes cross-reactive across all coronaviruses (PAN library control) than younger participants, suggesting that either older participants have more exposure to seasonal coronaviruses over their lifetime, or, more likely, that the immune responses of older patients target a broader repertoire of epitopes, a factor which is associated with control of infection [33].

Surprisingly, we found no correlations of increased specific humoral or cellular immunity with long-term complications. This is possibly because long-term complications following COVID-19 are probably caused by virus-specific pathophysiological changes, immunological dysregulation and inflammatory damage in response to the acute infection but independent from acute immune defense against the virus, which determines specific anti-SARS-CoV-2 antibody levels and T-cell counts [34]. Counter-intuitively, however, patients with persistent loss of smell or taste lasting at least 3 months had strikingly low levels of specific IgA. IgA is mainly responsible for mucosal immunity as the major proportion of neutralizing antibodies to SARS-CoV-2 in the early humoral immune response [28]. Given the efficient evasion of innate immune responses by SARS-CoV-2 [35], these long-term complications may not be due to an over-reactive immune response, but rather due to a lack or delay of specific humoral immunity at a critical phase or site of infection.

Our study clearly has limitations. Although our study provides an estimate of common long-term effects, it is not powered to detect rare complications, or new-onset disorders, due to low participant numbers. On the other hand, we were able to utilize a far broader range of assays in our smaller cohort than comparable studies for correlative analyses. Another limitation is the sex imbalance in our study, with a clear preponderance of male participants. This was due to the continuous inclusion of all patients with mild COVID-19 who had registered for a plasma donor and agreed to participate in the study without considering sex.

## 5. Conclusions

Taken together, we find no correlation of immunological factors with the development of long-term complications (long COVID) despite extensive immunological characterization of patients with exclusively mild infection, with the exception of a weak correlation of low IgA at early time points in participants that suffered loss of taste and smell. We show that antibody and T-cell responses correlate with the duration of infection, and that these are comparably stable between 6 and 12 months post-infection. Nevertheless, neutralizing antibodies were found in only 50% of participants at 12 months post-infection. Similarly, there was a significant decrease in cellular immunity from 6 to 12 months post-infection measured by ELISpot, even though 80% of participants developed measurable CoV-2-specific CD4+T-cells indicative of long-term memory.

## Figures and Tables

**Figure 1 jcm-10-03305-f001:**
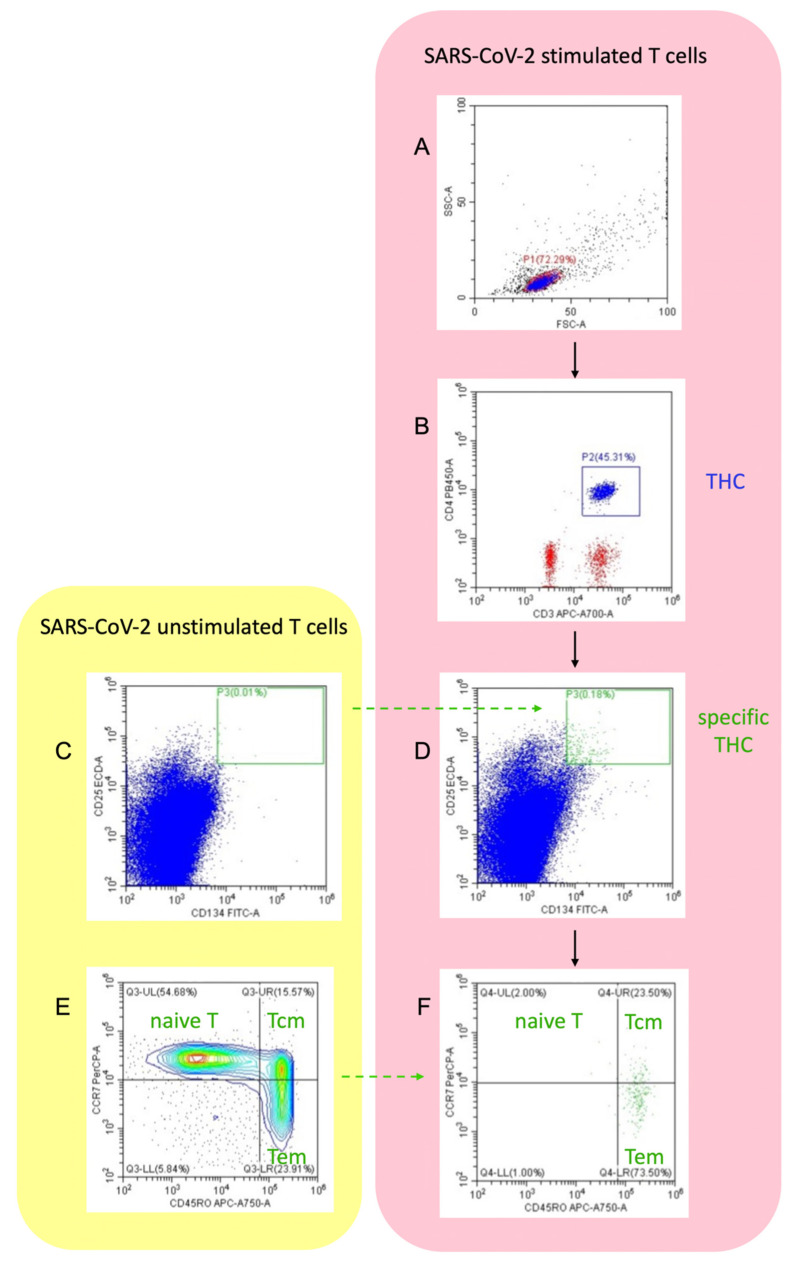
Flow cytometric gating strategy of AIM assay to identify specific T-helper cells (THC), and central memory T-cell (Tcm) and effector memory T-cell (Tem) subsets from a participant included in the study: (**A**) lymphocytes (Gate P1) were gated using forward and side scatter; (**B**) gated on P1: THC (Gate P2) were defined by co-expression of CD3 and CD4; (**D**) gated on P2: specific THC (Gate P3) were analyzed as CD25 high/CD134 high double-positive cell population after stimulation with SARS-CoV-2; (**C**) borders of Gate P3 were defined by means of unstimulated THC population; (**F**) gated on P3: specific THC were further divided into Tcm and Tem; (**E**) borders between Tcm, Tem and naïve T-cells were defined by unstimulated THC.

**Figure 2 jcm-10-03305-f002:**
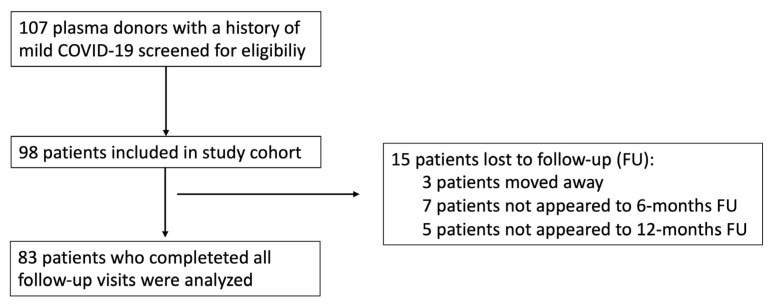
Consort diagram for this prospective study.

**Figure 3 jcm-10-03305-f003:**
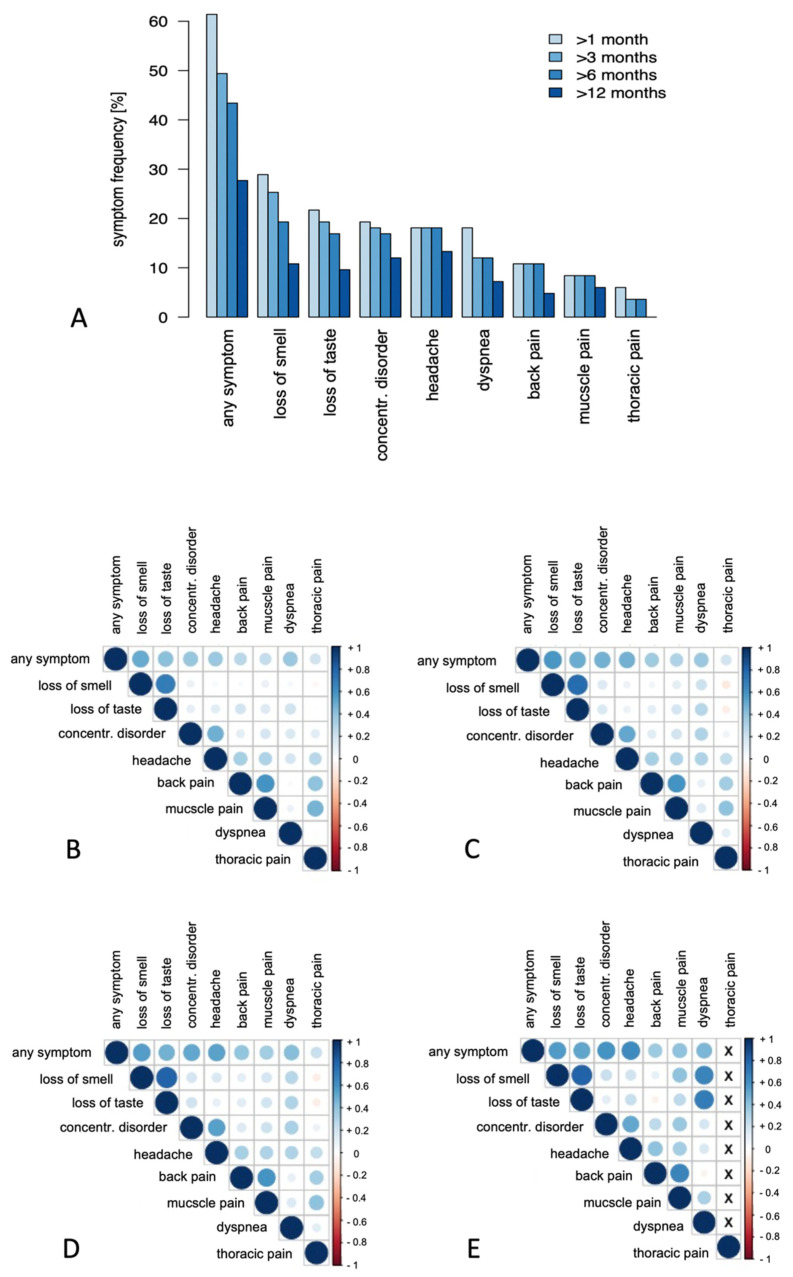
(**A**) Frequency of symptoms in patients after mild COVID-19 lasting at least 1, 3, 6 and 12 months, respectively, given in percentage. Correlation matrix displaying the co-occurrence of long-term symptoms for at least 1 (**B**), 3 (**C**), 6 (**D**) and 12 (**E**) months after COVID-19. Color intensity and the size of the circle are proportional to the correlation coefficients.

**Figure 4 jcm-10-03305-f004:**
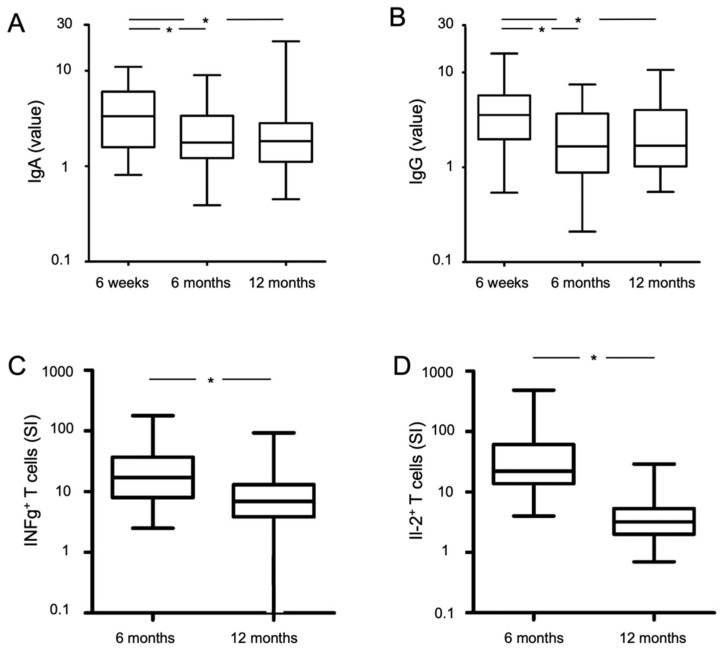
Course of IgA/IgG antibody level (**A**,**B**) and specific T-cells (**C**,**D**) over time. Black lines in boxplots represent median values. * *p* < 0.001 significant value. At 12-months measurement points, only non-vaccinated participants were considered.

**Figure 5 jcm-10-03305-f005:**
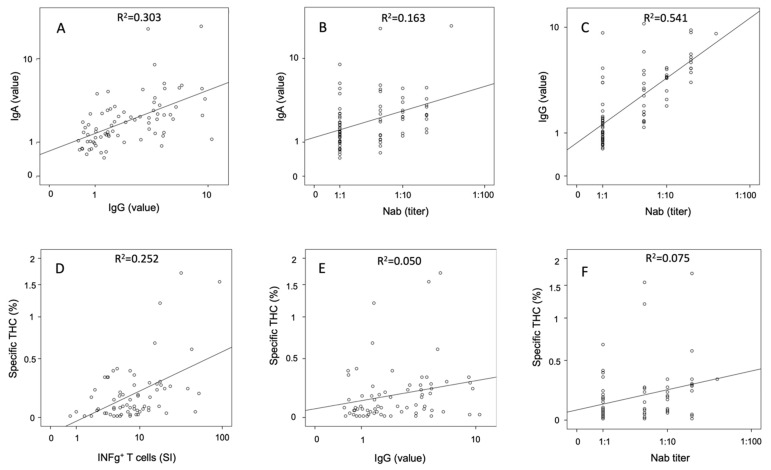
Correlation analyses of humoral and cellular immunity assays in non-vaccinated participants at 12-months follow-up: (**A**) IgG value vs. IgA value; (**B**) titer of Nab vs. IgA value; (**C**) titer of Nab vs. IgG value; (**D**) INFg^+^ T-cells vs. specific THC; (**E**) IgG value vs. specific THC; (**F**) titer of Nab vs. specific THC; Nab, neutralizing antibody; THC, T-helper cells; SI, stimulation index.

**Table 1 jcm-10-03305-t001:** Demographics and characteristics of 83 evaluable patients during acute COVID-19. * Homogeneous distribution across the years.

Demographics and Characteristics	COVID-19 Patients
Age in years; median(range)	42 (19–62) *
Gender	
male; n	63 (76%)
female; n	20 (24%)
Symptoms during acute COVID-19	
fever (>38 °C); n	49 (59%)
cough; n	32 (39%)
loss of smell; n	25 (30%)
loss of taste; n	18 (22%)
headache; n	15 (18%)
dyspnoe; n	15 (18%)
myalgia; n	15 (18%)
sore throat; n	11 (13%)
pain of the joints; n	10 (12%)
rhinits; n	8 (9%)
diarrhoe; n	6 (7%)
nausea; n	1 (1%)
Duration of symptoms (days); median (range)	11 (1–35)
Patient’s care during acute COVID-19	
non-hospitalization; n	80 (96%)
hospitalization; n	3 (4%)
Time period (days in median (range)) from	
onset of COVID-19 to 6-week visit	37 (26–99)
onset of COVID-19 to 6-month visit	204 (184–280)
onset of COVID-19 to 1-year visit	372 (348–407)
Pre-existing co-morbidities	
hypothyroidism	4 (5%)
asthma	2 (2%)
hay fever	4 (5%)

**Table 2 jcm-10-03305-t002:** Analysis of humoral immunity against SARS-CoV-2 at 6-week and 6-month follow-up (all participants non-vaccinated) as well as at 12-month follow-up grouped between vaccinated and non-vaccinated participants. Results are given as median values (range): Nab, neutralizing antibodies; *, at least 14 days before 12 months follow-up visit; p1 value, 6-week vs. 6-month; p2 value, vaccinated * vs. non-vaccinated at 12-months visit.

Assay	6-Week Visit (n = 83)	6-Month Visit (n = 83)	p1 Value	12-Month Visit (Non-Vaccinated, n = 77)	12-Month Visit (Vaccinated *, n = 5)	p2 Value
IgA value	3.38 (0.8–11.0)	1.80 (0.4–9.0)	<0.001	1.9 (0.5–20.3)	22.3 (10.3–22.3)	<0.001
IgG value	3.65 (0.5–15.8)	1.80 (0.2–7.5)	<0.001	1.7 (0.6–10.6)	35.0 (13.1–40.8)	<0.001
Nab titer	1:5 (1:1–1:640)	not done		1:1 (1:1–1:40)	1:320 (1:160–1:320)	<0.001

**Table 3 jcm-10-03305-t003:** Analysis of cellular immunity against SARS-CoV-2 measured by AIM and ELISpot at 6- and 12-month follow-up. Results are given as median values (range). ^#^ Values determined for participants with detectable specific T-helper cells (THC) only; INFg, interferon gamma; Il-2, interleukin-2; SI, stimulation index; Tcm, central memory T-helper cells; Tem, effector memory T-helper cells; *, Vaccination at least 14 days before; p1 value, 6-week vs. 6-month visit; p2 value, vaccinated vs. non-vaccinated at 12-month visit.

Assay		6-Month Visit		12-Month Visit (Non-Vaccinated)		p1 Value	12-Month Visit (Vaccinated *)		p2 Value
Elispot	INFg SI value	n = 51	19.0 (2.5–179)	n = 76	6.9 (0.0–93)	<0.001	n = 5	22.5 (5.0–74)	0.032
	Il-2 SI value	n = 32	22.0 (4.0–485)	n = 70	3.2 (0.7–29)	<0.001	n = 5	4.5 (2.2–19)	0.206
	Double positive SI	n = 32	6.8 (0.0–74)	n = 70	3.3 (0.0–29)	0.001	n = 5	2.5 (1.0–17)	0.781
AIM	Specific THC (%)			n = 70	0.08 (0.01–1.71)		n = 5	0.1 (0.02–0.93)	0.845
	THC SI value			n = 70	5.1 (1.0–53)		n = 5	10.0 (2.0–16)	0.462
	specific Tcm (%) ^#^			n = 56	39 (8–64)		n = 5	53 (23–100)	0.080
	specific Tem (%) ^#^			n = 56	58 (20–84)		n = 5	44 (1–67)	0.039
	ratio Tem / Tcm ^#^			n = 56	1.5 (0.3–10.5)		n = 5	0.8 (0.01–2.9)	0.059

## Data Availability

Data are publicly viewable.

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
