# Peer review of "One Year after Mild COVID-19: The Majority of Patients Maintain Specific Immunity, But One in Four Still Suffer from Long-Term Symptoms"

_jcm, 2021, doi:10.3390/jcm10153305_

Round 1
Reviewer 1 Report
Rank et al provide useful analysis of COVID-19 patient immunity and long-term symptoms. This data is valuable, yet additional analysis and significant additions to the discussion are needed.
- The introduction should provide more details into what currently is and is not known regarding the immune response (specifically the duration of the IgA and IgG levels and T cell memory) so it is clearer what novel information this manuscript is describing. Some recent articles examining this topic include: https://www.frontiersin.org/articles/10.3389/fimmu.2021.636768/full
https://www.nature.com/articles/s41577-021-00550-x
- The Materials and Methods should include your experimental details so they could be replicated if needed (rather than referencing another paper). Include the details of what the AIM assay is measuring.
- In Figure 1 it is unclear where C and E fit into the gating strategy, please clarify this in the figure and the figure legend.
- In Figures 4 and 5 it would be valuable to have a clear comparison of experimental values for infected patients compared to the control patients over time.
- Include the R squared values in figure 5 so the reader does not have to go to the supplemental figure to know that.
- In the discussion include a comparison of your long-term immune memory (IgA, IgG, helper T cells) findings to those of other studies that have recently been published. Is yours similar or different? Provide context on your findings for the reader.
- Have a typo on line 283 (CoV-2-pecific).
- Provide possible reasons for there being no correlation of immunological factors and the development of long-term complications. What could be the reason for this?
Reviewer 2 Report
This is a small but interesting study that I think is worth publishing with some revision. While the authors acknowledge several limitation of their study, the overwhelmingly male cohort (76% male) is not discussed (see comments below). The demographic of the vaccinated cohort is also not discussed (does it differ from the demographic of the 12 month unvaccinated cohort??)
Would suggest the following edits:
- Line 41 "Any persistent complications..." would remove the word ANY at the beginning of the sentence.
- Recommend indicating in the method section the number of vaccinated patients and when they were vaccinated. They were excluded from the 12 month data but no mention if they were vaccinated at 6 months and their data excluded or not at the 6 month point. This data would be important to mention.
- Line#94... word "performed" should be replace with word PREFORMED.
- This study is overwhelmingly a male cohort. This needs to be made amply clear from the get go that the number of male outweigh the number of female in the study. Data should be analyzed to see if there are significant differences in terms of antibody and T cell responses between male and female in this study-with statistic provided if possible. If this is not possible, it should be made clear throughout the text that differences between male and females may exists in terms of immunological responses but they have not been captured by the study and this should be acknowledged as a major limitation of this study.
- The age aspect is addressed to some extent in the conclusion however since the age is 19-62 one wonders if it is equally distributed or the study includes mostly young vs old individuals. Age is an important factor in terms of immunological parameters and breaking down age distribution and nothing if there are significant differences between old vs young individuals should be indicated when that data can be calculated (based on cohort size for young vs old) especially for antibodies (some data is discussed for T cells).
- Figure 4 A and B, suggest changing scale to lower max values so it is easier to visualize (maybe max of 40 or 50 with scale in increments of 10...) or slightly different scale but something that would enable easier visualization of the data.
- Figure 5, trend lines are heavily influenced in some graphs by outliers. One may consider redoing the analysis by removing statistical outliers. Would include the r2 on each graph for ease of interpretation rather than in supplementary table.
- The vaccinated data needs to be checked to make sure the vaccination group is demographically comparable to non-vaccinated (in terms of age and sex). If statistical demographic differences exists between the vaccinated and non-vaccinated 12 month cohort this should be acknowledge in the study. Readers need to be informed if the demographics are different and may have affected the results. We need to know if we are comparing apples to oranges.
Round 2
Reviewer 1 Report
Nice study that will contribute to the field